# UniINR: Unifying Spatial-Temporal INR for RS Video Correction, Deblur, and Interpolation with an Event Camera

## Abstract

Images captured by rolling shutter (RS) cameras under fast camera motion often contain obvious image distortions and blur, which can be modeled as a row-wise combination of a sequence of global shutter (GS) frames within the exposure time. Naturally, recovering high-frame-rate GS sharp frames from an RS blur image needs to simultaneously consider RS correction, deblur, and frame interpolation. Tacking this task is nontrivial, and to the best of our knowledge, no feasible solutions exist by far. A naive way is to decompose the whole process into separate tasks and simply cascade existing methods; however, this results in cumulative errors and noticeable artifacts. Event cameras enjoy many advantages, *e.g.*, high temporal resolution, making them potential for our problem. To this end, we propose the **first** and novel approach, named **UniINR**, to recover arbitrary frame-rate sharp GS frames from an RS blur image and paired event data. Our key idea is *unifying spatial-temporal implicit neural representation (INR) to directly map the position and time coordinates to RGB values to address the interlocking degradations in the image restoration process*. Specifically, we introduce spatial-temporal implicit encoding (STE) to convert an RS blur image and events into a spatial-temporal representation (STR). To query a specific sharp frame (GS or RS), we embed the exposure time into STR and decode the embedded features pixel-by-pixel to recover a sharp frame. Our method features a lightweight model with only $0.379M$ parameters, and it also enjoys high inference efficiency, achieving $2.83ms/frame$ in $31\times$ frame interpolation of an RS blur frame. Extensive experiments show that our method significantly outperforms prior methods.

## 1 Introduction

Most consumer-level cameras based on CMOS sensors rely on a rolling shutter (RS) mechanism. These cameras dominate the market owing to their benefits, *e.g.*, low power consumption (Janesick et al., 2009). In contrast to the global shutter (GS) cameras, RS cameras capture pixels row by row; thus, the captured images often suffer from obvious spatial distortions (*e.g.*, stretch) and blur under fast camera/scene motion. It has been shown that naively neglecting the RS effect often hampers the performance in many real-world applications (Hedborg et al., 2012; Lao & Ait-Aider, 2020; Zhong et al., 2021; Zhou et al., 2022). In theory, an RS image can be formulated as a row-wise combination of sequential GS frames within the exposure time (Fan & Dai, 2021; Fan et al., 2023).

In this regard, it is meaningful to *recover high-frame-rate sharp GS frames from a single RS blur image* as the restored high-frame-rate sharp GS frames can directly facilitate many downstream tasks in practice. Intuitively, achieving this goal often requires simultaneously considering RS correction, deblurring, and frame interpolation. However, tackling this task is nontrivial because multiple degradations, such as RS distortion, motion blur, and temporal discontinuity (Meilland et al., 2013; Su & Heidrich, 2015), often co-exist for CMOS cameras (Zhong et al., 2021). The co-existence of various image degradations complicates the whole GS frame restoration process. ***To the best of our knowledge, no practical solutions exist in the literature to date***. A naive way is to decompose the whole process as separate tasks and simply cascading existing image enhancement networks can result in cumulative errors and noticeable artifacts. For example, a simple consideration of cascading a frame interpolation network (Bao et al., 2019) with an RS correction network produces degraded results, as previously verified in (Naor et al., 2022).

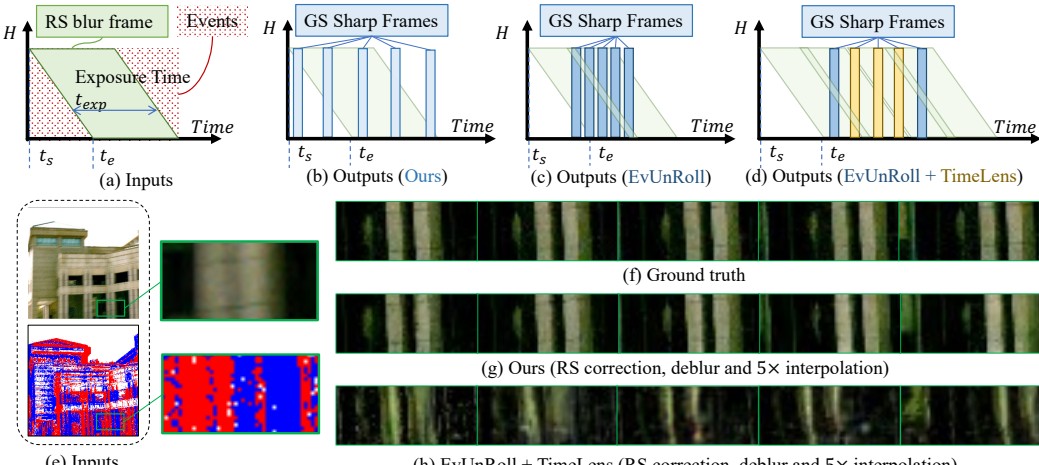

Figure 1: Inputs and the outputs of our method, EvUnRoll, and EvUnRoll+TimeLens. Inputs are shown in (a), which includes an RS blur image and events. $t_s$ and $t_e$ are the start and end timestamps of RS, and $t_{exp}$ is the exposure time. Our outputs are shown in (b), which is a sequence of GS sharp frames during the whole exposure time of the RS blur image. (c) shows outputs of EvUnRoll, which can only recover the GS sharp frames in a limited time instead of the whole exposure time of the RS blur frame. (d) shows outputs of EvUnRoll+TimeLens. *More details are in Sec. C7 in Supp. Mat.*

Event cameras offer several advantages, such as high-temporal resolution, which make them suitable for various image restoration tasks (Wang et al., 2020; Zhou et al., 2022; Tulyakov et al., 2021; Song et al., 2023; 2022). eSL-Net (Wang et al., 2020) proposes an event-guided sparse learning framework to simultaneously achieve image super-resolution, denoising, and deblurring. TimeLens (Tulyakov et al., 2021) integrates a synthesis-based branch with a warp-based branch to boost the performance of the video frame interpolation. DeblurSR (Song et al., 2023) and E-CIR (Song et al., 2022) take advantage of the high temporal resolution of events by converting a blurry frame into a time-to-intensity function, using spike representation and Lagrange polynomials, respectively. EvUnRoll (Zhou et al., 2022) leverages events as guidance to enhance RS correction by accounting for nonlinear motion during the desired timestamp. ***However, these methods focus on either deburring or RS correction and can not recover arbitrary frame-rate sharp GS frames from a single RS blur image.*** An example is depicted in Fig. 1 (h), showing that simply cascading event-guided RS correction model (*e.g.*, EvUnroll (Zhou et al., 2022)) and interpolation model (*e.g.*, TimeLens (Tulyakov et al., 2021)) to recover high-frame-rate sharp GS frames results in obvious artifacts.

In this paper, we make the **first** attempt to propose a novel yet efficient learning framework, dubbed **UniINR**, that can ***recover arbitrary frame-rate sharp GS frames from an RS blur image and events***. Our key idea is to learn a spatial-temporal *implicit neural representation (INR) to directly map the position and time coordinates to RGB values to address the co-existence of degradations in the image restoration process*. This makes it possible to exploit the spatial-temporal relationships from the inputs to achieve RS correction, deblur, and interpolation simultaneously. One distinct advantage of our method is that it is relatively lightweight with only **0.379$M$** parameters. We formulate the task —recovering high-frame-rate sharp GS frames from an RS blur image and paired event data —as a novel *estimation* problem, defined as a function, $F(\boldsymbol{x}, t, \theta)$. Here, $\boldsymbol{x}$ denotes the pixel position $(x, y)$ of an image, $t$ denotes the timestamp during the exposure time, and $\theta$ denotes the function's parameters. Our proposed framework consists of three parts: spatial-temporal implicit encoding (STE), exposure time embedding (ETE), and pixel-by-pixel decoding (PPD). Specifically, STE first utilizes sparse learning-based techniques (Wang et al., 2020) to extract a spatial-temporal representation (STR) $\theta$ from events and an RS blur image (Sec. 3.2.1). To query a specific sharp frame of RS or GS pattern, we then model the exposure information as a temporal tensor $T$ in ETE (Sec. 3.2.2). Finally, PPD leverages an MLP to decode sharp frames from the STR and the temporal tensor $T$ (Sec. 3.2.3), allowing for the generation of a sharp frame at any given exposure pattern (*e.g.*, RS or GS). One notable advantage of our approach is its ***high efficiency***, as it only requires using the STE once, regardless of the number of interpolation frames. In practice, as frame interpolation multiples rise from $1\times$ to $31\times$, the time taken increases from $31ms$ to $86ms$. Thus, at $31\times$ interpolation, each frame's processing time is merely 2.8ms, whereas the cascading approach (EvUnRoll + TimeLens) requires more than $177ms$ (Sec. 4.2).

We conduct a thorough evaluation of our proposed method, including both quantitative and qualitative analyses, using a higher resolution ($256 \times 256$) dataset than that of the previous methods ($180 \times 240$) (Song et al., 2023; 2022). Extensive experimental results demonstrate that our approach outperforms existing methods in RS correction, deblur, and interpolation (An example can be found in Fig. 1 (h)).

## 2 RELATED WORKS

### 2.1 EVENT-GUIDED IMAGE/VIDEO RESTORATION

**Event-guided Deblurring** Owing to the high temporal resolution afforded by events, prior studies (Sun et al., 2022; Wang et al., 2020; Shang et al., 2021; Kim et al., 2022) have incorporated events into the task of deblurring. These works focus on the reconstruction of a single GS sharp frame from the GS blur frame, guided by event data. The work most analogous to ours is EvUnroll (Zhou et al., 2022), which first leverages event cameras for RS correction, leveraging their low latency benefits. Nonetheless, EvUnroll primarily focuses on RS correction, with its optional deblurring module equipped to handle minor motion blur and reconstruct a sharp frame at the midpoint of the exposure time.

**Event-guided Deblurring + Interpolation** These studies can be bifurcated based on the quantity of input GS blur frames: single GS frame (Xu et al., 2021; Song et al., 2022; 2023; Haoyu et al., 2020) or multiple GS frames (Pan et al., 2019; Zhang & Yu, 2022; Lin et al., 2020). The former, such as E-CIR (Song et al., 2022) and DeblurSR (Song et al., 2023), convert a GS blur frame into a time-to-intensity function while the latter, *e.g.*, EDI (Pan et al., 2019), LEDVDI (Lin et al., 2020), and EVDI (Zhang & Yu, 2022) are both built upon the event-based double integral model (Pan et al., 2019). However, these methods primarily target GS frames affected by motion blur, leading to performance degradation when dealing with spatially distorted and RS blur frames.

Recently, a contemporaneous study (Zhang et al., 2023) also focused on RS Correction, Deblur, and VFI. However, this research primarily concentrated on the individual performance of a single model across the three tasks, without extensive experimentation or investigation into handling all three tasks concurrently. This constitutes the most significant distinction from our method.

### 2.2 FRAME-BASED VIDEO RESTORATION FOR RS INPUTS

**RS Correction + Interpolation** RSSR (Fan & Dai, 2021; Fan et al., 2023) is the first work that generates multiple GS frames from two consecutive RS frames by introducing bi-directional undistortion flows. CVR (Fan et al., 2022) estimates two latent GS frames from two consecutive RS frames, followed by motion enhancement and contextual aggregation before generating final GS frames.

**RS Correction + Deblurring** JCD (Zhong et al., 2021) proposes the first pipeline that employs warping and deblurring branches to effectively address the RS distortion and motion blur. However, JCD's motion estimation module, built upon the assumption of linear motion derived from DeepUnrollNet (Liu et al., 2020), encounters a significant performance degradation in real-world scenarios involving non-linear motion (Zhou et al., 2022). To eliminate the dependence of motion estimation, (Wang et al., 2022b) proposes a method that turns the RS correction into a rectification problem, which allows all pixels to start exposure simultaneously and end exposure line by line. *Differently, our method can recover arbitrary GS sharp frames during the exposure time of RS blur frames without the assumption of linear motion.*

### 2.3 IMPLICIT NEURAL REPRESENTATION (INR)

INR (Wang et al., 2021; Sitzmann et al., 2020; Chen et al., 2021; 2022; Lu et al., 2023) is proposed for parameterized signals (images, video or audio) in the coordinate-based representation, inspiring some researchers to explore the potential of INR in low-level vision tasks. LIIF (Chen et al., 2021) represents images as high-dimensional tensors and allows for upsampling at any scale through interpolation and decoding, followed by VideoINR (Chen et al., 2022), which extends LIIF to videos, enabling temporal and spatial upsampling at any scale. EG-VSR (Lu et al., 2023) incorporates events into the learning of INR to achieve random-scale video super-resolution. *Differently, we propose STE to directly map the position and time coordinates to RGB values to address the co-existence of degradations in the image restoration process. Our STE makes it possible to exploit the spatial-temporal relationships from the inputs to achieve RS correction, deblur, and interpolation simultaneously.*

## 3 METHODOLOGY

### 3.1 PROBLEM DEFINITION AND ANALYSIS

We formulate the task —*recovering arbitrary frame-rate sharp GS frames from an RS blur image and paired event data* —as a novel estimation problem, defined as a function, $F(\boldsymbol{x}, t, \theta)$. Here, $\boldsymbol{x}$ denotes

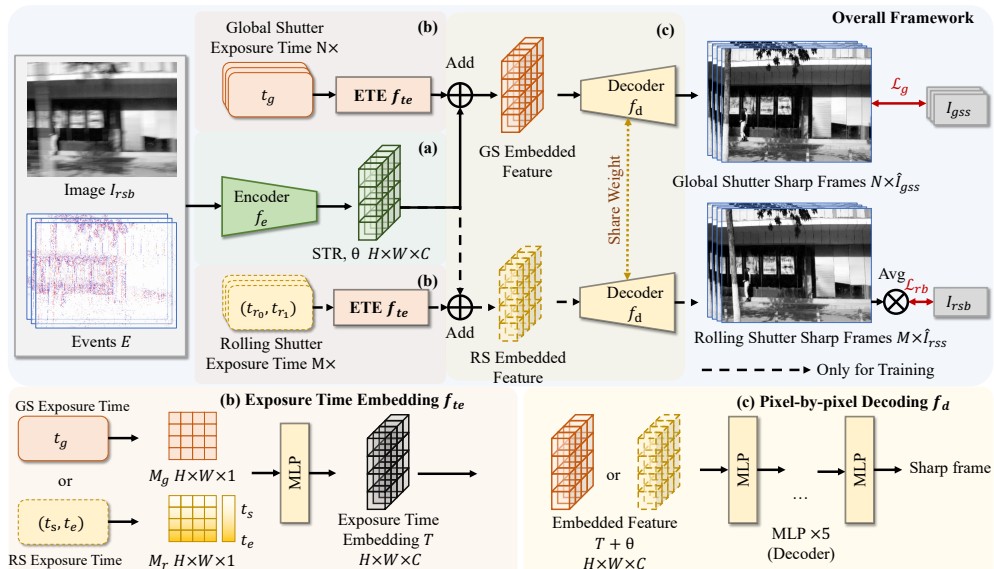

Figure 2: An overview of our framework, which consists of three parts, **(a)** the Spatial-Temporal Implicit Encoding (STE), **(b)** Exposure Time Embedding (ETE), and **(c)** Pixel-by-pixel decoding (PPD). Details of STE, ETE, and PPD are described in Sec. 3.2.1, Sec. 3.2.2, and Sec. 3.2.3. The inputs are an RS blur image $I_{rsb}$ and events, and the outputs are a sequence of GS frames and RS frames. RS frames are predicted only in training.

the pixel position $(x, y)$ of an image with a resolution of $H \times W$, $t$ denotes the timestamp during the exposure time, and $\theta$ denotes the parameters. The intuition behind this formulation is that there exists a relationship between the RS blur/sharp frame and the GS blur/sharp frame. We now describe it.

By defining a function $F(\boldsymbol{x}, t, \theta)$ mapping the pixel position $\boldsymbol{x} = (x, y)$ and timestamp $t$ to intensity or RGB value, we can obtain a GS sharp frame by inputting the desired timestamp $\hat{t}$ during the exposure time to the function, which can be formulated as:

$$I_{g,\hat{t}} = F(\boldsymbol{x}, \hat{t}, \theta) \tag{1}$$

As an RS image can be formulated as a row-wise combination of sequential GS frames within the exposure time (Fan & Dai, 2021; Fan et al., 2023), we can assemble an RS sharp frame $I_{r,t_s,t_e}$ from a sequence of GS sharp frames row by row given the RS start time $t_s$ and the end time $t_e$. That is, the $h$-th row of an RS frame is the same as the $h$-th row of a GS frame at $t_s^h$, and the exposure start timestamp of the $h$-th row of an RS frame is $t_s^h = t_s + h \times (t_e - t_s)/H$. Therefore, we can formally describe an RS sharp frame as follows:

$$I_{r,t_s,t_e} = \left\{ F\left(\boldsymbol{x}, t_s^h, \theta\right)[h], h \in [0, H] \right\}. \tag{2}$$

In principle, a blur frame can be regarded as the temporal average of a sequence of sharp frames (Nah et al., 2017; Zhang et al., 2020). Thus, a GS blur frame $I_{g,t_g,t_{exp}}$, where $t_g$ is the exposure start timestamp and $t_{exp}$ is the exposure time, can be expressed as the average of a sequence of GS sharp frames during the exposure time $t_{exp}$, which can be formulated as:

$$I_{g,t,t_{exp}} = \frac{1}{t_{exp}} \int_t^{t+t_{exp}} F(\boldsymbol{x}, t, \theta) dt \approx \frac{1}{N} \sum_{i=0}^{N} I_{g,t_0+i \times t_{exp}/N}, \tag{3}$$

where $N$ is the length of the GS frame sequence.

With the above formulation, an RS blur frame $I_{r,t_s \to t_e, t_{exp}}$ can thus be described based on the RS start time $t_s$, RS end time $t_e$, and exposure time of each scan line $t_{exp}$, as depicted in Fig. 1 (a). According to Eq. 2 and Eq. 3, the $h$-th row of an RS blur frame can be described as the temporal average of the $h$-th row in a sequence of GS sharp frames, which can be written as follows:

$$
\begin{aligned}
I_{r,t_s \to t_e, t_{exp}} &= \left\{ \frac{1}{t_{exp}} \int_{t_s^h}^{t_s^h+t_{exp}} F\left(\boldsymbol{x}, t, \theta\right)[h] dt, h \in [0, H] \right\} \\
&\approx \left\{ \frac{1}{N} \sum_{i=0}^{N} I_{g,t_s+i \times t_{exp}/N}[h], h \in [0, H] \right\}.
\end{aligned} \tag{4}
$$

An event stream $E$ consists of a set of event $e = (x, y, t, p)$, where each event is triggered and recorded with the polarity $p$ when the logarithmic brightness change at pixel $(x, y)$ exceeds a certain

threshold $C$, which can be approximated as the differential of $F(\boldsymbol{x}, t, \theta)$ with respect to the time dimension. *For details about the principle of event cameras, refer to the Suppl. Mat.*

To use event data $E$ as guidance, we need to address three challenges to estimate the mapping function $F(\boldsymbol{x}, t, \theta)$: **1)** how to find a function $f_e$ to encode the input RS blur image and events to $\theta$ of the mapping function $F(\boldsymbol{x}, t, \theta)$; **2)** how to find a function $f_{te}$ to represent the exposure information of desired RS or GS sharp frames as $t$ of the mapping function $F(\boldsymbol{x}, t, \theta)$; **3)** how to find a function $f_d$ to eliminate the need to input position information of desired RS or GS sharp frames as $p$ of the mapping function $F(\boldsymbol{x}, t, \theta)$. Therefore, our goal is to estimate $f_e$, $f_{te}$, and $f_d$ in order to get a mapped result, which can be formulated as:

$$I = F(\boldsymbol{x}, t, \theta) = F(\boldsymbol{x}, t, f_e(E, I_{rsb})) = F(\boldsymbol{x}, f_{te}(t), f_e(E, I_{rsb})) = f_d(f_{te}(t), f_e(E, I_{rsb})). \quad (5)$$

In the following section, we describe our framework based on Eq. 5 by substantiating $f_e$, $f_{te}$, and $f_d$.

## 3.2 PROPOSED FRAMEWORK

An overview of our UniINR framework is depicted in Fig. 2, which takes an RS blur image $I_{rsb}$ and paired events $E$ as inputs and outputs $N$ sharp GS frames $\{I_{gss}\}_{i=0}^N$ with a high-frame-rate. To substantiate the defined functions $f_e$, $f_{te}$, and $f_d$, as mentioned in Sec. 3.1, our proposed framework consists of three components: **1)** Spatial-Temporal Implicit Encoding (STE), **2)** Exposure Time Embedding (ETE), and **3)** Pixel-by-pixel Decoding (PPD). Specifically, we first introduce an STE with deformable convolution (Wang et al., 2022a) to encode the RS blur frame and events into a spatial-temporal representation (STR) (Sec. 3.2.1). To provide exposure temporal information for STR, we embed the exposure start timestamp of each pixel from the GS or RS by ETE. (Sec. 3.2.2). Lastly, the PDD module adds ETE to STR to generate RS or GS sharp frames (Sec. 3.2.3). We now describe these components in detail.

### 3.2.1 SPATIAL-TEMPORAL IMPLICIT ENCODING (STE)

Based on the analysis in Sec. 3.1, we conclude that the RS blur frame $I_{rsb}$ and events $E$ collectively encompass the comprehensive spatial-temporal information during the exposure process. In this section, we aim to extract a spatial-temporal implicit representation $\theta$ that can effectively capture the spatial-temporal information from the RS blur frame $I_{rsb}$ and events $E$.

To achieve this, we need to consider two key factors: (1) extracting features for the multi-task purpose and (2) estimating motion information. For the first factor, we draw inspiration from eSL-Net (Wang et al., 2020), which effectively utilizes events to simultaneously handle deblur, denoise, and super-resolution tasks. Accordingly, we design a sparse-learning-based backbone for the encoder. Regarding the second factor, previous works (Fan & Dai, 2021; Fan et al., 2022; 2023) commonly use optical flow for motion estimation in RS correction and interpolation tasks. However, optical flow estimation is computationally demanding (Gehrig et al., 2021; Zhu et al., 2019; Sun et al., 2018), making it challenging to incorporate it into the multiple task framework for RS cameras due to the complex degradation process. As an efficient alternative, we employ deformable convolution (Wang et al., 2022a) in our encoder to replace the optical flow estimation module. We adopt a 3D tensor with a shape of $H \times W \times C$ as the STR $\theta$, which can effectively address the interlocking degradations encountered in the image restoration process with a sparse-learning-based backbone and deformable convolution, as formulated as $\theta = f_e(E, I_{rsb})$ in Eq. 5. *More details in the Suppl. Mat.*

### 3.2.2 EXPOSURE TIME EMBEDDING (ETE)

As depicted in Fig. 2 (b), the primary objective of the ETE module is to incorporate the exposure time of either a rolling shutter (RS) frame $(t_s, t_e)$ or a global shutter (GS) frame $(t_g)$ by employing an MLP layer, resulting in the generation of a temporal tensor $T$. To achieve this, we design an ETE module, denoted as $f_{te}$, which takes the GS exposure time $t_g$ as input and produces the GS temporal tensor $T_g = f_{te}(t_g)$. Similarly, for RS frames, $T_r = f_{te}(t_{r_s}, t_{r_e})$ represents the RS temporal tensor, which is only used in training. The process begins by converting the exposure process information into a timestamp map, with a shape of $H \times W \times 1$. Subsequently, the timestamp map is embedded by increasing its dimensionality to match the shape of the STR. This embedding procedure allows for the integration of the exposure time information into the STR representation. We now explain the construction of timestamp maps for both GS and RS frames and describe the embedding method employed in our approach.

**GS Timestamp Map:** In GS sharp frames, all pixels are exposed simultaneously, resulting in the same exposure timestamps for pixels in different positions. Given a GS exposure timestamp $t_g$, the GS timestamp map $M_g$ can be represented as $M_g[h][w] = t_g$, where $h$ and $w$ denote the row and column indices, respectively.

**RS Timestamp Map:** According to the analysis in Sec. 3.1, pixels in RS frames are exposed line by line, and pixels in different rows have different exposure start timestamps. Given RS exposure information with start time $t_s$ and RS end time $t_e$, the RS timestamp map can be represented as $M_r[h][w] = t_s + (t_e - t_s) \times h/H$, where $h$, $w$, $H$ denote the row and column indices and height of the image, respectively.

**Time Embedding:** The timestamp maps, $M_r$ and $M_g$, represent the timestamps of each pixel in a specific frame (RS or GS) with a shape of $H \times W \times 1$. However, the timestamp map is a high-frequency variable and can pose challenges for learning neural networks (Vaswani et al., 2017). Some approaches (Vaswani et al., 2017; Wang et al., 2021) propose a combination function of sine and cosine to encode the positional embedding. Nonetheless, calculating the derivative of the positional embedding is difficult, limiting its practical application to image enhancement tasks. In this paper, we utilize a one-layer MLP to increase the dimension for embedding. The whole embedding process is formulated as $T_g = f_{te}(t_g)$ for GS frames, and $T_r = f_{te}(t_{r_s}, t_{r_e})$ for RS frames, as depicted in Fig. 2(b). The MLP consists of a single layer that maps the timestamp map $M_r$ or $M_g$ to the same dimension $H \times W \times C$ as the spatial-temporal representation (STR) $\theta$, as described in Sec. 3.2.1.

### 3.2.3 PIXEL-BY-PIXEL DECODING (PPD)

As shown in Fig. 2 (c), the goal of PPD is to efficiently query a sharp frame from STR $\theta$ by the temporal tensor $T$. It is important that the encoder is invoked only once for $N$ times interpolation, while the decoder is called $N$ times. Therefore, the efficiency of this query is crucial for the overall performance. The query's inputs $\theta$ capture the global spatial-temporal information, and $T$ captures the temporal information of the sharp frame (GS or RS). Inspired by previous works (Mildenhall et al., 2021; Chen et al., 2021), we directly incorporate the temporal tensor $T$ into the STR $\theta$ to obtain an embedded feature with a shape of $H \times W \times C$ for each query. This additional embedded feature combines the global spatial-temporal information with the local exposure information, enabling straightforward decoding to obtain a sharp frame. To avoid the need for explicit positional queries, we employ a pixel-by-pixel decoder. The decoder, denoted as $f_d$ in Eq. 5, employs a simple 5-layer MLP $f_{mlp}^{\circlearrowleft^5}$ architecture. The reconstructed output $I$ after decoding can be described in Eq. 6, where $\oplus$ means element-wise addition.

$$I = f_d(f_{te}(t), f_e(E, I_{rsb})) = f_d(T, \theta) = f_{mlp}^{\circlearrowleft^5}(T \oplus \theta). \tag{6}$$

### 3.2.4 LOSS FUNCTION

**RS Blur Image-guided Integral Loss:** Inspired by EVDI (Zhang & Yu, 2022), we formulate the relationship between RS blur frames and RS sharp frames. Given a sequence of RS sharp frames generated from the decoder, the input RS blur frame $I_{rsb} = \frac{1}{M} \sum_{i=1}^{M} (\hat{I}_{rss}^i)$, where $M$ represents the length of the RS image sequence. In this way, we can formulate the blur frame guidance integral loss between the reconstructed RS blur frame and the original RS blur frame as $\mathcal{L}_b = \mathcal{L}_c(\hat{I}_{rsb}, I_{rsb})$, where $\mathcal{L}_c$ indicates *Charbonnier loss* (Lai et al., 2018).

**Total Loss:** Apart from RS blur image-guided integral loss $\mathcal{L}_b$, we incorporate a reconstruction loss $\mathcal{L}_{re}$ to supervise the reconstructed GS sharp frames. Our method consists of two losses: RS blur image-guided integral loss and the reconstruction loss, where $\lambda_b, \lambda_{re}$ denote the weights of each loss:

$$\mathcal{L} = \lambda_b \mathcal{L}_b + \lambda_{re} \mathcal{L}_{re} = \lambda_b \mathcal{L}_c(\hat{I}_{rsb}, I_{rsb}) + \lambda_{re} \frac{1}{N} \sum_{k=1}^{N} \mathcal{L}_c(\hat{I}_{gss}^k, I_{gss}^k). \tag{7}$$

## 4 EXPERIMENTS

**Implementation Details:** We utilize the Adam optimizer (Kingma & Ba, 2014) for all experiments, with learning rates of $1e - 4$ for both Gev-RS (Zhou et al., 2022) and Fastec-RS (Liu et al., 2020) datasets. Using two NVIDIA RTX A5000 GPU cards, we train our framework across 400 epochs with a batch size of two. In addition, we use the mixed precision (Micikevicius et al., 2017) training tool provided by PyTorch (Paszke et al., 2017), which can speed up our training and reduce memory usage. PSNR and SSIM (Wang et al., 2004) are used to evaluate the reconstructed results.

**Datasets: 1) Gev-RS dataset (Zhou et al., 2022)** features GS videos at $1280 \times 720$, 5700 fps. We reconstruct such frames and events from the original videos, downsampling to $260 \times 346$ (Scheerlinck et al., 2019). Events and RS blur frames are synthesized using vid2e (Gehrig et al., 2020). We adopt EvUnroll's (Zhou et al., 2022) 20/9 train/test split. **2) Fastec-RS dataset (Liu et al., 2020)** offers GS videos at $640 \times 480$, 2400 fps. We apply identical settings for resizing, event creation, and RS blur. The dataset is split into 56 training and 20 testing sequences. **3) Real-world dataset (Zhou et al., 2022)** is the only available real dataset, containing four videos with paired RS frames and events. Due to the lack of ground truth, it offers only quantitative visualizations. *More details in Suppl. Mat..*

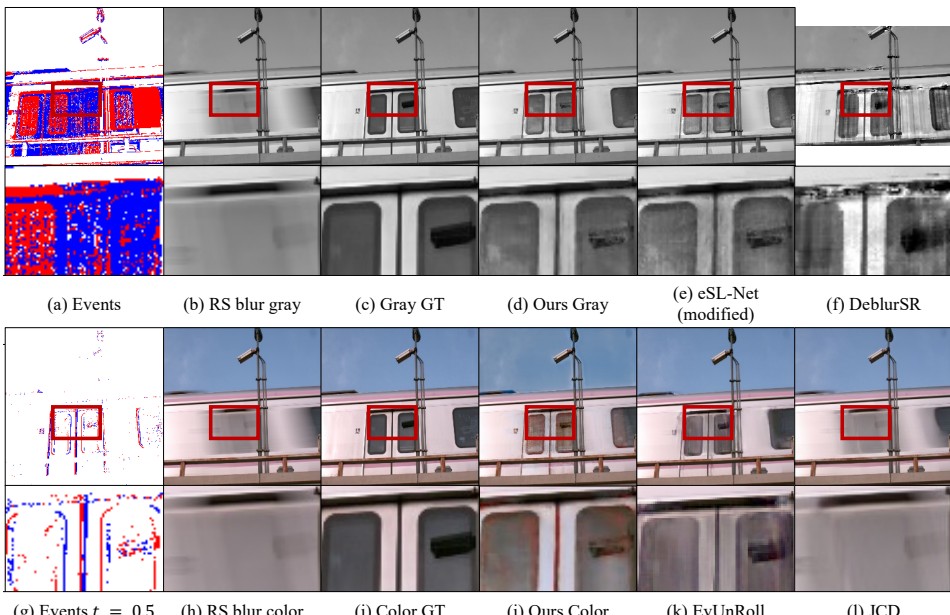

Figure 3: Visual Comparisons on RS correction and deblurring on Gev-RS (Zhou et al., 2022) dataset. The image resolution of DeblurSR (Song et al., 2023) is $180 \times 240$.

## 4.1 COMPARISON WITH SOTA METHODS

Our experiments are conducted on both simulated and real datasets. While the simulated dataset enables us to obtain accurate quantitative results, evaluating on the real dataset offers insights into the generation ability of our method.

We compare our methods with recent methods with two different settings in these two datasets: **(I)** the experiment with a single GS sharp frame result, including JCD (Zhong et al., 2021) (frame-based RS correction and deblurring), EvUnroll (Zhou et al., 2022) (event-guided RS correction) and eSL-Net (Wang et al., 2020) (event-guided deblurring). **(II)** the experiment with a sequence of GS sharp frames result, which includes DeblurSR (Song et al., 2023) (event-guided deblurring and interpolation), and the combination of EvUnroll (Zhou et al., 2022) and TimeLens (Tulyakov et al., 2021) (event-guided video frame interpolation). In addition, we test our model's generation ability by comparing it with EvUnRoll (Zhou et al., 2022) using real data. While this real data is solely reserved for testing, both our model and EvUnRoll are trained on the simulation dataset. *More explanations of setting (II) are in Supp. Mat..*

We evaluate JCD, EvUnroll, TimeLens, and DeblurSR with the released code. We modified eSL-Net by adjusting its parameterization initialization method and removing the up-sampling module, allowing it to be well trained on our datasets. The outputs of eSL-Net and DeblurSR are grayscale frames, and the outputs of JCD, EvUnroll, and the combination of EvUnroll and TimeLens are RGB frames. For fairness, our network is trained with the input of grayscale and RGB images, respectively.

The quantitative results for experiments generating a single GS sharp frame ($1\times$) and those producing a sequence of GS sharp frames ($3\times, 5\times, 9\times$) are presented in Tab. 1. In comparison to methods that yield a single GS sharp frame, our approach exhibits remarkable performance in both gray and RGB frames, surpassing the best-performing methods (eSL-Net (Wang et al., 2020) in gray and EvUnroll (Zhou et al., 2022) in RGB) by **1.48dB** and **4.17dB** on the Gev-RS (Zhou et al., 2022) dataset, respectively. In scenarios where a sequence of GS sharp frames is produced, our method attains optimal performance for both gray and RGB frames, achieving an increase of up to **13.47dB** and **8.49dB** compared to DeblurSR (Song et al., 2023) and EvUnroll (Zhou et al., 2022)+Time-Lens (Tulyakov et al., 2021) on the Gev-RS (Zhou et al., 2022) dataset, respectively. The substantial performance decline of DeblurSR (Song et al., 2023) can be ascribed to the interdependence between RS correction and deblur. The performance reduction of EvUnroll+TimeLens can be accounted for by the accumulation of errors arising from this cascading network, as shown in Fig. 1(h).

The qualitative results, as depicted in Fig. 11, showcase the effectiveness of our proposed method on both grayscale and RGB inputs. These results demonstrate our approach's ability to generate

Table 1: Quantitative results for RS correction, deblurring, and frame interpolation. TL refers to TimeLens Tulyakov et al. (2021). EU refers to EvUnroll Zhou et al. (2022). eSL-Net* represents a modified model based on eSL-Net Wang et al. (2020).

| | Methods | Inputs | | | Gev-RS | | Fastec-RS | |
|---|---|---|---|---|---|---|---|---|
| | | Frame | Event | Params(M) ↓ | PSNR ↑ | SSIM ↑ | PSNR ↑ | SSIM ↑ |
| 1× | eSL-Net* | 1 gray | ✓ | 0.1360 | 31.64 | 0.9614 | 32.45 | 0.9186 |
| | UniINR (Ours) | 1 gray | ✓ | 0.3790 | **33.12** | **0.9881** | **34.62** | **0.9390** |
| | JCD | 3 color | ✗ | 7.1659 | 18.59 | 0.5781 | 21.31 | 0.6150 |
| | EU | 1 color | ✓ | 20.83 | 26.18 | 0.8606 | 29.76 | 0.8693 |
| | UniINR (Ours) | 1 color | ✓ | 0.3792 | **30.35** | **0.9714** | **33.64** | **0.9299** |
| 3× | DeblurSR | 1 gray | ✓ | 21.2954 | 17.64 | 0.554 | 21.17 | 0.5816 |
| | UniINR (Ours) | 1 gray | ✓ | 0.3790 | **31.11** | **0.9738** | **33.23** | **0.9210** |
| | EU + TL | 2 color | ✓ | 93.03 | 21.86 | 0.7057 | 24.81 | 0.7179 |
| | UniINR (Ours) | 1 color | ✓ | 0.3792 | **28.36** | **0.9348** | **32.72** | **0.9147** |
| 5× | DeblurSR | 1 gray | ✓ | 21.2954 | 18.35 | 0.6107 | 22.86 | 0.6562 |
| | UniINR (Ours) | 1 gray | ✓ | 0.3790 | **30.84** | **0.9673** | **32.82** | **0.9147** |
| | EU + TL | 2 color | ✓ | 93.03 | 21.59 | 0.6964 | 24.46 | 0.7140 |
| | UniINR (Ours) | 1 color | ✓ | 0.3792 | **28.41** | **0.9062** | **32.13** | **0.9053** |
| 9× | DeblurSR | 1 gray | ✓ | 21.2954 | 18.86 | 0.6502 | 23.96 | 0.7049 |
| | UniINR (Ours) | 1 gray | ✓ | 0.3790 | **30.54** | **0.9579** | **32.21** | **0.9051** |
| | EU + TL | 2 color | ✓ | 93.03 | 21.24 | 0.6869 | 23.99 | 0.7029 |
| | UniINR (Ours) | 1 color | ✓ | 0.3792 | **27.21** | **0.8869** | **29.31** | **0.8590** |

Table 2: Quantitative comparison in PSNR, SSIM, and LPIPS on EvUnRoll simulation dataset (Zhou et al., 2022). The numerical results of DSUN, JCD, and EvUnRoll are provided by (Zhou et al., 2022).

| Method | Frames | Event | Params(M) ↓ | PSNR ↑ | SSIM ↑ | LPIPS ↓ |
|---|---|---|---|---|---|---|
| DSUN (Liu et al., 2020) | 2 | ✗ | 3.91 | 23.10 | 0.70 | 0.166 |
| JCD (Zhong et al., 2021) | 3 | ✗ | 7.16 | 24.90 | 0.82 | 0.105 |
| EvUnRoll (Zhou et al., 2022) | 1 | ✓ | 20.83 | 30.14 | 0.91 | 0.061 |
| UniINR(Ours) | 1 | ✓ | **0.38** | **30.61** | **0.9285** | **0.048** |

sharp frames devoid of RS distortion, yielding the most visually pleasing outcomes in challenging scenarios involving a fast-moving train with motion blur and RS distortion. Comparatively, the results of eSL-Net and EvUnroll exhibit discernible noise, particularly evident around the train door within the red region of Fig. 11. Another approach, JCD, falls short in recovering sharp frames within such complex scenes. This failure can be attributed to the insufficient availability of frame-based methods which rely on the assumption of linear motion. Furthermore, the results obtained using DebluSR (Song et al., 2023) display noticeable artifacts, particularly in the context of the moving train. These artifacts hinder satisfactory frame reconstruction in such dynamic environments.

*Bad case analysis:* The color distortion in Fig. 11 (j) can be attributed to the insufficient color information in the challenging scene of a fast-moving train. From the input (Fig. 11 (h)), it can be noticed that the degree of motion blur is extremely severe and the blurry frame cannot provide valid color information. Furthermore, according to the principle of the generation of the event, the event is triggered by intensity change and it cannot provide color information.

**EvUnRoll Simulation Dataset:** To achieve a more equitable comparison with EvUnRoll, we evaluate our method on the simulated dataset employed by EvUnRoll, shown in Tab. 2. It's important to emphasize that the dataset includes paired data consisting of RS blur, RS sharp, and GS sharp. For our model's training, we specifically utilize the paired images of RS blur and GS sharp. As a one-stage approach, our method directly transforms an RS-blurred image into a GS-sharp image avoiding accumulated error, and thus has better performance.

**Real-world Dataset:** Fig. 4 shows real-world results. The input frame exhibits rolling shutter distortions, such as curved palette edges. In contrast, events show global shutter traits. Both our method and EvUnRoll correct these distortions effectively. Due to the lack of ground truth, quantitative analysis is not possible. Notably, our method avoids artifacts and errors, outperforming EvUnRoll in palette scenarios. *For further discussion please refer to the Supp. Mat..*

## 4.2 ABLATION AND ANALYTICAL STUDIES

**Importance of Exposure Time Embedding:** We conduct the experiments to evaluate the impact of learning-based position embedding, with a comparative analysis to sinusoid position embed-

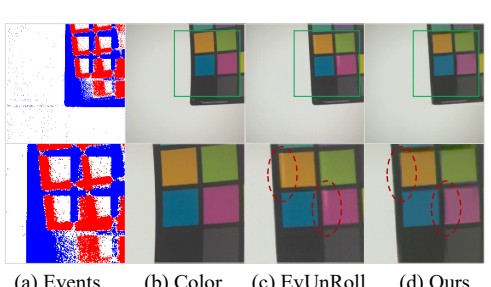

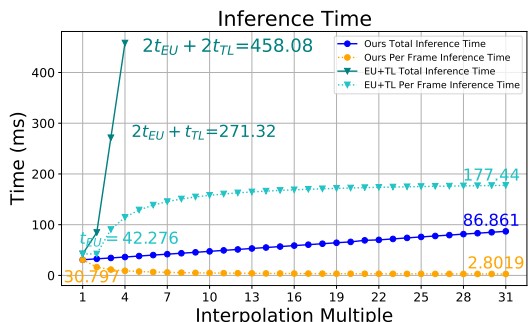

(a) Events     (b) Color     (c) EvUnRoll     (d) Ours

Figure 4: Visualization results in a real-world dataset (Zhou et al., 2022). (a) is the events visualization results. (b) are the input RGB images that have clear rolling shutter distortions. (c) is the output of EvUnRoll. (d) are the outputs of our method. The red circle in (c) has color distortion.

Figure 5: Comparison of inference time of our method with EvUnroll + TimeLens. $t_{EU}$ and $t_{TL}$ represent the respective inference times of EvUnRoll and TimeLens. The axes represent frame interpolation multiples ($1\times$ to $31\times$) and time. $2T_{EU}$ and $2t_{TL}$ means calling EvUnRoll twice and TimeLens twice.

ding (Vaswani et al., 2017). As indicated in Tab. 3, learning-based position embedding outperforms sinusoid position embedding, with advancements of up to **1.11dB** on average. This superior efficacy is attributable to the intrinsic adaptability of the learning-based position embedding.

**Importance of RS Blur Image-guided Integral Loss:** The effectiveness of the RS blur image-guided integral Loss across diverse interpolation settings is depicted in Tab. 4. The findings point towards the enhancement in PSNR for high interpolation configurations (*e.g.*, $9\times$) upon employing this loss.

**Inference Speed:** Fig. 5 shows our method's inference time across $1\times$ to $31\times$ interpolation. The total time rises modestly, e.g., from $30.8\ ms$ at $1\times$ to $86.9\ ms$ at $31\times$, a 2.8-fold increase for a 31-fold interpolation. The average frame time even decreases at higher multiples, reaching $2.8\ ms$ at $31\times$. Compared to EvUnRoll (Zhou et al., 2022) and TimeLens (Tulyakov et al., 2021), our method is more computationally efficient, requiring only 72% of EvUnRoll's $42.3\ ms$ for RS correction and deblurring. For $N$-fold frame insertion using EvUnRoll + TimeLens, EvUnRoll is counted twice, and TimeLens $N - 2$ times. This advantage is amplified in high-magnification scenarios, where TimeLens costs $186.76\ ms$ per call. Our calculations focus on GPU time, excluding data I/O, which further increases EvUnRoll and TimeLens' time consumption. *More discussions are in Supp. Mat..*

Table 3: Ablation for learning-based position embedding.

|  | Position Embedding | PSNR | SSIM |
|---|---|---|---|
| $1\times$ | Sinusoid | 32.46 | 0.9851 |
|  | Learning | **33.12** | **0.9881** |
| $3\times$ | Sinusoid | 30.83 | 0.9723 |
|  | Learning | **31.11** | **0.9738** |
| $5\times$ | Sinusoid | 30.70 | **0.9678** |
|  | Learning | **30.84** | 0.9673 |
| $9\times$ | Sinusoid | 30.51 | 0.9560 |
|  | Learning | **30.54** | **0.9579** |
|  |  | **+1.11** | **+0.0059** |

Table 4: Ablation for the loss function.

|  | $\mathcal{L}_b$ | PSNR | SSIM |
|---|---|---|---|
| $1\times$ | ✗ | 33.12 | 0.9881 |
|  | ✓ | 33.14 | 0.9844 |
| $3\times$ | ✗ | 31.11 | 0.9738 |
|  | ✓ | 31.09 | 0.9768 |
| $5\times$ | ✗ | 30.84 | 0.9673 |
|  | ✓ | 30.83 | 0.9784 |
| $9\times$ | ✗ | 30.54 | 0.9579 |
|  | ✓ | 30.61 | 0.9538 |
|  |  | **+0.060** | **+0.0063** |

## 5 CONCLUSION

This paper presented a novel approach that simultaneously uses events to guide rolling shutter frame correction, deblur, and interpolation. Unlike previous network structures that can only address one or two image enhancement tasks, our method incorporated all three tasks concurrently, providing potential for future expansion into areas such as image and video super-resolution and denoising. Furthermore, our approach demonstrated high efficiency in computational complexity and model size. Regardless of the number of frames involved in interpolation, our method only requires a single call to the encoder, and the model size is a mere 0.379M.

**Limitations** Our analysis utilizes simulated data and real-world datasets, the latter of which lacks ground truth. Acquiring real data with ground truth is challenging. In future work, we aim to address this limitation by employing optical instruments, such as spectroscopes, to obtain real-world data with ground truth for quantitative evaluation.

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
