# OpenReview forum: "UniINR: Unifying Spatial-Temporal INR for RS Video Correction, Deblur, and Interpolation with an Event Camera"
_ICLR.cc/2024/Conference — Submitted to ICLR 2024_

### Official Review · Reviewer_gAyr · 2023-10-29

**Soundness:** 3 good
**Presentation:** 3 good
**Contribution:** 2 fair
**Rating:** 5
**Confidence:** 4

**Summary:**

This paper aims to recover arbitrary frame-rate sharp GS frames from an RS blur image and events. To this end, the authors introduce the spatial-temporal implicit neural representation (INR) to recover the sharp GS frames by mapping the position and time coordinates to RGB values. The input RS blur image and events are fused to construct a spatio-temporal representation and extract features from them at any time and position. This method has a lightweight and fast network structure, and also achieves high image reconstruction performance.

**Strengths:**

1. The authors thoroughly explain the framework and the proposed INR representation is experimentally verified for its ability to represent spatio-temporal features.
2. This method has a lightweight and fast network structure, and also achieves high image reconstruction performance.

**Weaknesses:**

1. The fusion of events and images is completely end-to-end, with no explicit connection between the exposure time and the timestamp in events. This looks like a whole black box and lacks interpretability. There are established methods that associate exposure time, optical flow, and events, which is more intuitive, but of course introduces more computation, which is a key claim of this method.
2. Visual comparisons on real dataset does not show significant differences. Are there more obvious samples in the dataset? Otherwise I don't think it can support the generalizability of the model.
3. In C.5 of the Supplementary Material, the author mentions that concat can perform better than addition, so why does the author continue to use only addition?

**Questions:**

The authors have done a lot of explanations and experiments and have also fully demonstrated the effectiveness of the INR representation for RS deblurring and corr. However, I don't see much new from the model design. This is neither the first application of the INR representation to event-based vision, nor is it an exciting one. My opinion is borderline reject.

---

> ### Author Response · Authors · 2023-11-21
> **Response to Reviewer gAyr**
>
> Dear Reviewer, gAyr,
>
> First and foremost, we would like to express our sincere gratitude for the time and effort you have invested in reviewing our manuscript. Your insightful comments and suggestions are immensely valuable to us. We have carefully considered each point you raised and provided detailed responses below to address your concerns.
>
> ---
> ## Weakness Response
>
> ### W1.1: NO explicit connection between the exposure time and the timestamp in events.
> We apologize for any confusion caused by our presentation.
> To clarify, we do indeed model the relationship between exposure time and the timestamp of events in our paper (Section 3.1, Lines 133-169).
> Specifically, Equations 3 and 4 in our paper introduce the connection between exposure time and timestamps. Additionally, the role of the event in this formulation is detailed from Line 158 to Line 160.
> This approach is the linkage between the eve timestamp of events and the image exposure time.
>
> ### W1.2: Associate exposure time, optical flow, and events, which is more intuitive.
> Furthermore, our INR-based method utilizes Exposure Time Embedding to incorporate time information into the INR. This allows us to directly obtain intensity values at any given time, thereby circumventing the need for more complex operations like optical flow computation. As you pointed out, while such methods are more intuitive, they also introduce significantly more computational complexity. Our approach achieves comparable results with fewer parameters, aligning with our goal of computational efficiency.
>
> ### W2: More obvious samples in the dataset.
> Thank you for pointing out the need for more distinct examples in our dataset. We acknowledge your concerns regarding the visual comparisons on the real dataset. To address this, we direct your attention to Figure 4 (c) in our paper, where the results from EvUnroll exhibit noticeable artifacts, as indicated by the red circles. These artifacts underscore the differences and improvements our method offers.
>
> Furthermore, we have included additional real-world data visualizations in the supplementary materials (Figure 6). These examples effectively demonstrate our method's ability to correct deformations, particularly in challenging scenarios such as outdoor windows. This robust performance in diverse real-world conditions highlights the generalizability and effectiveness of our rolling shutter correction approach.
>
> ### W3: Concat and addition operate:
> Thank you for your insightful query regarding our choice of operation in our model.
> Indeed, while concatenation can offer marginal improvements, it comes at the cost of increased model size and computational overhead. In our experiments, we observed that the performance differences between addition, multiplication, and concatenation were relatively minimal, with variations around 0.03 dB and 0.04 dB, which are not substantial enough to justify the additional complexity.
>
> Given our emphasis on maintaining a lightweight and efficient design, we decided to continue using addition, which aligns with our objectives of computational efficiency and simplicity. However, recognizing the potential benefits of alternative operations, we plan to include these options in our publicly released code. This will allow users to explore and leverage different operation types according to their specific requirements and constraints.

---

> > ### Author Response · Authors · 2023-11-21
> > **Response to Reviewer gAyr**
> >
> > ## Question Response
> >
> > ### Q1: New from the model design:
> > We appreciate your concern regarding the novelty of our model design and would like to highlight two key aspects of innovation: the overall framework and the temporal embedding.
> >
> > **Novelty of the Framework:** The structure of our model introduces a significant shift in the approach to RS correction. Unlike previous methods such as EvUNRoll, which typically involve a two-step process of deblurring followed by RS correction, our method consolidates these steps into a single, unified process. This integration reduces cumulative errors, a critical advantage especially in scenarios with high degrees of blur, as evidenced in Table 1 of our main paper. This streamlined process not only enhances the accuracy but also improves the overall efficiency of the method.
> >
> > **Novelty of Temporal Embedding:** The primary innovation in our approach is the utilization of temporal embedding as opposed to optical flow, which has been prevalent in prior RS correction and INR methods. Optical flow, while effective, tends to add substantial complexity to the model. By leveraging temporal embedding, our method simplifies the model architecture, making it more compact and efficient. This approach not only eases the computational burden but also boosts performance. We believe this represents a substantial departure from traditional methods of integrating time-related information into INR for RS deblurring and correction. This novel integration offers a fresh perspective in the field of event-based vision systems and sets our work apart from previous applications of INR in this area.
> >
> > In light of these clarifications and the unique contributions of our work, we respectfully request that you might reconsider the scoring of our submission.

---

### Official Review · Reviewer_QpjK · 2023-10-30

**Soundness:** 2 fair
**Presentation:** 3 good
**Contribution:** 3 good
**Rating:** 5
**Confidence:** 1

**Summary:**

In this paper, an event-based RS video correction, deblur, and interpolation method is proposed. Different from existing methods that decompose the whole process into multiple tasks, the authors take advantage of the high temporal resolution of event cameras to recover arbitrary frame-rate sharp GS frames from an RS blur image and paired event data. The key idea is unifying spatial-temporal implicit neural representation (INR) to directly map the position and time coordinates to RGB values to address the interlocking degradations in the image restoration process. Several experiments have been conducted to demonstrate the validity of the proposed method.

**Strengths:**

1.	The question of using event cameras for unified RS image enhancement makes sense, although event cameras have been explored on separate tasks. This is the first attempt.
2.	To this end, the authors propose a series of modules to perform the proposed task

**Weaknesses:**

1. Although the question raised is forward-looking, the technical details given by the authors are not very innovative, like some stacking of existing models. This makes it harder for reviewers to capture what the real contribution are. It is recommended that authors summarize the contribution visibly.
2. A more important issue is that of experimental validation. Overall experimental validation is not sufficient to prove the validity of the proposed methodology. The authors mention the use of two different experimental settings:
 (I) the single GS sharp frame restoration, i.e., the RS correction and deblurring. However, the EvUnroll is only the event-guided RS correction method, and eSL-Net is only the event-guided deblurring method. The target algorithm that the authors should be comparing should be correction+deblurring simultaneously. Thus, it is recommended that the authors use a combination of correction + deblurring algorithms for experimental comparisons, including but not limited to EvUnroll +eSL-Net, there are many event-guided deblurring algorithms available.
(II) a sequence of GS sharp frames restoration, i.e., the RS correction, deblurring and interpolation. However, the DeblurSR is the event-guided deblurring and interpolation method and does not target the correction. Therefore, the experimental comparison is not very fair. Meanwhile, the combination of EvUnroll and TimeLens does not target the deblurring.
Overall, the authors are advised to conduct a fairer and more comprehensive experimental comparison
3. Whether the algorithms being compared are using pre-trained models or re-trained? Reviewers noted that the compared methods were significantly lower than the proposed methods.

**Questions:**

See Weaknesses.

---

> ### Author Response · Authors · 2023-11-21
> **Response to Reviewer QpjK**
>
> Dear QpjK:
> First and foremost, we would like to express our sincere gratitude for the time and effort you have invested in reviewing our manuscript. Your insightful comments and suggestions are immensely valuable to us. We have carefully considered each point you raised and provided detailed responses below to address your concerns.
>
>
> ---
>
> ### W1: Summarization of Contributions
> We are grateful for your feedback and take this opportunity to succinctly restate the key contributions of our work, earnestly anticipating your recognition of these aspects:
>
> **Novel Research Question:** This study pioneers the recovery of high-frame-rate sharp GS frames at arbitrary frame rates from RS blur images and paired event data. Our approach diverges fundamentally from traditional methods, charting a new course in event-based imaging. This first-of-its-kind effort brings a fresh perspective to the field.
>
> **Innovative Methodology:** We have developed a one-stage framework utilizing a unified implicit neural representation (INR). Distinctively, it maps space-time coordinates to RGB values through an Exposure Time Embedding module coupled with a Pixel-by-pixel Decoding module. This framework eschews the conventional reliance on optical flow, traditionally known for its computational complexity. Our innovative integration of temporal and spatial data sets our method apart from existing approaches.
>
> **Superior Performance:** Our model demonstrates exceptional performance across datasets with varying degrees of blur, as evidenced in Tables 1 and 2 of the main paper. It consistently outperforms the existing method, EvUNRoll, while maintaining a remarkably efficient architecture with only 0.379M parameters. This efficiency, combined with its effectiveness, underscores our model's superiority over more complex models.
>
>
> ### W2: Experimental Validation
> We value your thoughtful feedback regarding our experimental validation. To clarify and address your concerns:
>
> **Single GS Sharp Frame Restoration:**
>
> - **EvUnroll:** As stated in EvUnroll's documentation, it includes an optional image deblurring module. By processing the RS image first through this deblurring module and then the RS correction module, EvUnroll effectively accomplishes both deblurring and RS correction in sequence.
> - **eSL-Net:** For our specific experimental needs, we have adapted eSL-Net by modifying its parameter initialization method and removing the up-sampling module. These adjustments ensure its compatibility and efficient functioning in our experimental framework.
>
> **Sequence of GS Sharp Frames Restoration:**
> - **Combination of EvUnroll and TimeLens:** This combination is capable of handling deblurring, RS correction, and interpolation. The process involves sequentially feeding inputs through EvUnroll’s optional deblurring module, then its RS correction module, and finally, TimeLens’s interpolation network. For a detailed depiction of this procedure, we refer you to **Fig. 10 (II)** in the supplementary material, which offers a comprehensive illustration of the entire process.
>
> We trust these detailed explanations and methodological adjustments adequately address your concerns regarding the fairness and thoroughness of our experimental comparisons.
>
>
> ### W3: Whether the algorithms being compared are using pre-trained models or re-trained?
> Response: We appreciate your insightful observations regarding the training status of models in our comparative analysis. We recognize the importance of clear experimental descriptions and are committed to enhancing the clarity in the experimental section of our paper. To address your concerns, we offer the following clarifications:
>
> **Pre-trained Models or Re-trained?:**
> We utilized the pre-trained model of TimeLens owing to the unavailability of its training code.
> For all other models in our comparison, we re-trained them using their respective publicly released codes.
>
> **Fairness in Comparative Analysis:**
> To ensure an equitable comparison, we evaluated our method under the same settings as those used for EvUnroll.
> This comparison, detailed in Table 2 of our main paper, underscores the superior performance of our approach.
> It’s noteworthy that EvUnroll supports the simulated dataset used in Table 2.
>
> **Focus on larger degrees of Motion Blur and Distortion:**
> Benefiting from the one-stage method, our model has better performance for large degree motion blur.
> To this end, we created a simulated dataset with a larger degree of blur, as presented in Table 1.
> The parameters for this simulation are detailed in Supplementary Material C.1.
> As indicated in Table 1, the performance of the compared methods falls below that of our proposed method.
> This discrepancy can be attributed to the inherent challenges associated with effectively handling severe RS correction, motion deblurring, and frame interpolation simultaneously.

---

### Official Review · Reviewer_w92c · 2023-10-30

**Soundness:** 3 good
**Presentation:** 3 good
**Contribution:** 2 fair
**Rating:** 5
**Confidence:** 5

**Summary:**

This paper proposed a method named UniINR that interpolates arbitrary sharp global shutter (GS) frames from a single rolling shutter (RS) frame with the assistance of an event camera. The proposed model unifies the spatial-temporal implicit neural representation (INR) to address the interlocking degradations in the process of image restoration. Experimental results show that the proposed method outperforms other comparing methods in both quality evaluations and efficiency evaluations.

**Strengths:**

- The authors proposed the first novel yet efficient learning framework that can recover arbitrary frame-rate sharp GS frames from an RS blur image with corresponding event signals.

- The network is light-weighted with only 0.379M parameters, which is much smaller than EvUnroll model. Furthermore, the efficiency of this model is evident in its remarkably swift inference time, outperforming previous methods by a substantial margin.

- The performance of the proposed method is better than the state-of-the-art methods in both quantitative evaluations such as PSNR, SSIM, LPIPS, and visual results. The videos provided in the supplementary materials show the better performance of the proposed method.

**Weaknesses:**

- In the fusion of RS frames and event streams, there is no specific modules or design that focus on the domain gap between these two modalities. Since the format and information recorded by the intensity values and event signals are quite different.

- The event signals directly record the coordinates of events triggered at that time. Unlike the intensity frames, in one event voxel tensor, not all the pixels contain event signals. As a result, directly applying INR to event voxel tensor doesn’t consider the sparsity of event signals.

**Questions:**

- The comparison of the methods in this paper is a crucial aspect of evaluating its contributions. However, it has come to my attention that the performance of the compared methods in this submission does not appear to match the results presented in their original papers. It is imperative that a fair and comprehensive evaluation of the methods is conducted to ensure the validity of the comparisons and the reliability of the conclusions drawn in this work. Specifically, the performance of EvUnroll and JCD are worse than their original papers in some metrics and qualitative evaluations. Did the authors retrained their models in a different way?

- The authors claim for multiple times that they are the first to achieve arbitrary frame interpolation from an RS blur image and events. I think it is not that convincing. Because EvUnroll is also able to reconstruct multiple GS sharp frames.

- In the supplementary code, I found that in the SCN module, the input images and events are multiplied to get a feature called ‘x1’. I wonder what the meaning of multiplication is.

- In experimental part, this paper does not utilize optical flow for frame interpolation, unlike methods such as VideoINR [1] and MoTIF [2], which use INR to predict optical flow and then perform frame interpolation. However, in this paper, optical flow is not employed, and it seems that there is no dedicated comparison with optical flow methods in the conducted ablation experiments.

[1] Chen et al., VideoINR: Learning Video Implicit Neural Representation for Continuous Space-time Super-resolution, CVPR 2022.

[2] Chen et al., MoTIF: Learning Motion Trajectories with Local Implicit Neural Functions for Continuous Space-Time Video Super-Resolution, ICCV 2023.

---

> ### Author Response · Authors · 2023-11-21
> **Response to Reviewer w92c**
>
> Dear Reviewer w92c,
>
> First and foremost, we would like to express our sincere gratitude for the time and effort you have invested in reviewing our manuscript. Your insightful comments and suggestions are immensely valuable to us. We have carefully considered each point you raised and provide detailed responses below to address your concerns.
>
> ---
>
> ## Weaknesses
>
> ### W1: Specific modulesor design that focus on the domain gap.
> In response to your query about the fusion of RS frames and event streams, we would like to direct your attention to lines 181-198 of our paper, where we elaborate on the rationale behind our encoder's design. Our approach is influenced by the principles of sparse learning, which we've found to be particularly effective when bridging the domain gap between the two modalities. The sparse learning framework has been proven effective in previous work for handling the distinct formats and information characteristics inherent in intensity values and event signals. By integrating this framework into our encoder design, we effectively address the unique challenges posed by the fusion of these two distinct data types.
>
> ### W2: Directly applying INR to event voxel tensor doesn’t consider the sparsity of event signals.
> In addressing your concern regarding the application of INR to event voxel tensors, it's important to clarify that our INR framework operates on the features outputted by the Encoder, rather than directly on the event data. The Encoder is specifically designed to capitalize on the high temporal resolution of event signals, enabling it to learn a continuous spatiotemporal representation. This design choice effectively addresses the sparsity of event signals, ensuring that our INR framework is applied to a more contextually rich and temporally informed feature set, rather than directly to the sparse event data.

---

> > ### Author Response · Authors · 2023-11-21
> > **Response to Reviewer w92c**
> >
> > ## Question:
> >
> > ### Q1: Concerns About Method Comparisons
> > - a) **Reference to EvUnRoll's Performance in Table 2:** The performance data for EvUnRoll cited in Table 2 of our paper is derived from the simulated dataset provided by EvUnRoll itself.
> > This ensures that the comparison is made under similar conditions as the original study, thereby maintaining the integrity and reliability of the performance metrics presented.
> >
> > - b) **Dataset Differences in Table 1:** In Table 1, we employ a dataset simulated by our, which introduces a greater degree of blur compared to the dataset used in EvUnRoll's original study. This higher degree of blur complexity showcases the enhanced performance of our method. EvUnRoll separates deblurring and RS correction, tends to exhibit a rapid drop in performance when faced with larger degree blur levels. In contrast, our integrated method maintains robust performance under these challenging conditions. The specifics of these dataset differences and their implications are further elaborated in Supplementary Material C.1.
> >
> > - **c) Retraining EvUnRoll:** For the retraining of EvUnRoll, we completely followed the open-source code provided by its authors. This retraining process was conducted to ensure that our comparisons are as fair and accurate as possible, reflecting true-to-life performance metrics.
> >
> > ### Q2: Arbitrary frame interpolation:
> > We appreciate the opportunity to clarify this aspect of our work. While EvUnroll is indeed capable of interpolating frames, its capability is not as flexible as our method in terms of generating frames at arbitrary timestamps. Our approach uniquely enables the interpolation of frames at any specified time point, which is a distinct advancement over EvUnroll's capabilities. This key difference is demonstrated in Figure 1 of our main paper and further elaborated in Figures 9 and 10 of the supplementary material. Our method’s ability to interpolate frames at any given time, irrespective of the original frame rate or timing, sets it apart from EvUnroll and underscores the novelty of our contribution in this area.
> >
> > ### Q3: Feature called ‘x1’.
> > Thank you for your inquiry regarding the multiplication operation in the SCN module within our supplementary code. This multiplication between input images and events to obtain the feature 'x1' is a key aspect of our sparse learning approach. This method is inspired by and closely references the techniques outlined in the previous paper on eSL (Event Sparse Learning). By multiplying the intensity frames with the event data, we effectively integrate the high temporal resolution information from the event signals with the spatial information from the intensity frames.
> > This integration is crucial for capturing the dynamic changes in the scene, a cornerstone of our method's ability to accurately interpret and process the event data in conjunction with traditional intensity frames.
> >
> > ### Q4: Compared to VideoINR and MoTIF:
> > Thank you for highlighting the differences in approach between our work and methods like VideoINR and MoTIF. We would like to clarify our methodology and its distinctions:
> >
> > **Different Research Focus:**
> > VideoINR and MoTIF are primarily focused on super-resolution (SR) and video frame interpolation (VFI), where warp and multi-frame fusion are applicable and effective. Our research, however, concentrates on addressing the challenges of blur and rolling shutter (RS) distortion. Due to the nature of these distortions, warp and multi-frame fusion, as used in VideoINR and MoTIF, are theoretically not feasible for our specific problem setting.
> >
> > **Unique Methodological Approach:**
> > The research problems we tackle necessitate fundamentally different methodologies compared to VideoINR. While VideoINR integrates SpatialINR and TemporalINR sequentially for frame interpolation and super-resolution, our method innovates by developing a unified INR strategy. This unified INR incorporates Exposure Time Embedding and Pixel-by-Pixel Decoding, enabling simultaneous RS correction, deblurring, and frame interpolation. This approach is specifically tailored to address the complexities of RS deformation and blur, an aspect not considered in VideoINR.
> > Our Exposure Time Embedding module is a crucial differentiator, allowing for the generation of both RS and GS images with specific exposure time information. This capability is unique to our approach and is not achievable by VideoINR, given their different focus.
> >
> > **Different Experimental Dataset:**
> > Owing to the divergent research objectives and methodologies, the dataset used in our study is distinct from that utilized in VideoINR. The choice of dataset in our research is aligned with our goals to specifically address RS correction, deblurring, and frame interpolation under conditions of RS distortion and blur.
> > By elaborating these key differences, we hope to provide a clearer understanding of our method's unique contributions and the rationale behind our experimental choices.

---

### Official Review · Reviewer_xnov · 2023-11-01

**Soundness:** 2 fair
**Presentation:** 3 good
**Contribution:** 2 fair
**Rating:** 5
**Confidence:** 4

**Summary:**

This paper proposes a method that recovers HFR and sharp global shutter frames from a rolling shutter blur image with the assistance of events. The authors propose an implicit neural representation to map the position and time coordinates to RGB image values. The experimental comparison with existing methods shows its effectiveness.

**Strengths:**

(1) Compared with the existing algorithms, the efficiency of the proposed algorithm is greatly improved.
(2) The experimental comparison shows that the proposed method exceeds the existing algorithms on both selected simulated and real dataset samples.

**Weaknesses:**

(1) A comprehensive performance comparison with EvShutter[1] is suggested, which has been published in CVPR2023.
(2) Both EvShutter and Evunroll demonstrate the performance of the proposed algorithms for eliminating the RS effect caused by fan rotation. To verify the robustness of the proposed method for different scenes, it is suggested to add similar experiments on real fan data.
(3) In Figure 5, it seems unfair and unnecessary to compare the running time of the proposed algorithm with the combination of EU and TL.
(4) The Evunroll also seems to take into account image deblurring and high frame-rate video output.

[1] Julius Erbach, Stepan Tulyakov, Patricia Vitoria, Alfredo Bochicchio, Yuanyou Li; Proceedings of the IEEE/CVF Conference on Computer Vision and Pattern Recognition (CVPR), 2023, pp. 13904-13913.

**Questions:**

Please see the weaknesses.

---

> ### Author Response · Authors · 2023-11-21
> **Response to Reviewer xnov**
>
> Dear Reviewer, xnov,
>
> First and foremost, we would like to express our sincere gratitude for the time and effort you have invested in reviewing our manuscript. Your insightful comments and suggestions are immensely valuable to us. We have carefully considered each point you raised and provided detailed responses below to address your concerns.
>
> ---
>
> ### W1: Comparison with EvShutter:
> We appreciate your suggestion regarding a comprehensive performance comparison with EvShutter. We would like to provide the following clarifications to address this concern:
>
> **Accessibility of the Dataset:** The RS-ERGB dataset proposed by EvShutter is not publicly available, limiting our ability to perform a direct comparison.
>
> **Opacity of Dataset Details:** The EvShutter paper does provide comparative results on the Fastec-RS dataset. However, essential details about this dataset, specifically the degree of simulation for Rolling Shutter and Blur, are not explicitly disclosed. This ambiguity impedes our ability to perform a precise and accurate comparative analysis with EvShutter.
>
> **Lack of Code and Implementation Details:** Another challenge is the unavailability of EvShutter's code. Additionally, the paper omits certain critical implementation details. These gaps pose a considerable obstacle in ensuring an accurate and equitable comparison.
>
> Despite these hurdles, we have endeavored to present a general comparison. In the Supplementary Material C.1 (lines 539-549), we describe our approach for synthesizing Rolling Shutter and Blur on the Fastec-RS dataset. Our experimental results demonstrate robust performance even under conditions of more pronounced blurring than what is seen with EvShutter (and EvUnRoll). Specifically, our model achieves a PSNR value of 33.64 and an SSIM value of 0.9299. These metrics significantly surpass the performance of 32.41 PSNR and 0.91 SSIM reported by EvShutter, highlighting the efficacy of our method under challenging conditions.
>
> ### W2: More Example:
> Thank you for suggesting additional experiments with real fan data to demonstrate the robustness of our method in different scenes. We acknowledge that EvUnRoll reported fan samples in their paper. However, the four real-world data scenes provided by EvUnRoll do not include these fan examples. In our experiments, we focused on these four scenarios, two indoor and two outdoor, to evaluate our method's effectiveness in diverse environments. The results of these experiments are showcased in Figure 6 of our supplementary materials. Our model demonstrates effective correction in both indoor and outdoor settings, with noticeable corrections observed especially in outdoor window scenes. This performance highlights our model's capability to adapt and correct various real-world distortions, reinforcing its robustness and versatility.

---

> > ### Author Response · Authors · 2023-11-21
> > **Response to Reviewer xnov**
> >
> > ### W3: Compare the running time with EU+TL:
> > We appreciate your observation regarding the comparison made in Figure 5. We understand the concern about the fairness and necessity of comparing our algorithm's running time with the combination of TL (TimeLens) and EvUnRoll. It's important to note that our method represents the first end-to-end solution for RS deblurring and frame interpolation (VFI), which presents a challenge in finding directly comparable methods.
> >
> > Given this context, we selected TL + EvUnRoll as the best available comparison, despite the differences in approach. This choice was made to provide a benchmark, albeit not a perfect one, against established methods in the field.
> >
> > To ensure the fairness of the speed test, as detailed in lines 334 to 342 of the main paper, we carefully avoided the inclusion of I/O times and focused solely on the running time on the GPU. This approach was taken to provide as fair and accurate a comparison as possible, given the constraints of available methods for comparison. Our goal was to offer a meaningful context for the efficiency of our end-to-end solution, even though a perfectly analogous method does not currently exist.
> >
> > ### W4:Deblurring and Interpolation in EvUnroll
> > We apologize for any confusion caused and acknowledge that EvUnroll does consider image deblurring and high frame-rate video output. However, it's important to note that EvUnroll is limited by its pretrained deblurring network, which only recovers RS clear images corresponding to the midpoint of the exposure time of the input RS blur image (as mentioned in Section 3.3 ‘Deblurring module’ of EvUnroll). Consequently, EvUnroll is only capable of reconstructing GS sharp frames at specific, limited periods. This limitation is depicted in Figure 1 (c) of our main paper and Figure 10 (I) of the supplementary material. In contrast, our method can reconstruct GS sharp frames at arbitrary timestamps, as shown in Figure 1 (b) of our main paper and Figure 9 of the supplementary material.

---

### Meta-Review · Area_Chair_wJAC · 2023-12-11

**Metareview:**

This paper proposes an effective method that uses events to guide rolling shutter frame correction, deblur, and interpolation. It introduces a STE to convert an RS blur image and events into a STR. It embeds the exposure time into STR and decodes the embedded features pixel-by-pixel to recover a sharp frame. Experimental results show the effect of the proposed methods.

All reviewers are not in favor of this manuscript due to the inappropriate experimental evaluations and limited technical contributions.

Based on the recommendations of reviewers, the paper is not ready for ICLR.

**Justification For Why Not Higher Score:**

N/A

**Justification For Why Not Lower Score:**

N/A

---

### Decision · Program_Chairs · 2024-01-16

Reject